# MedTrinity-25M: A Large-scale Multimodal Dataset with Multigranular Annotations for Medicine

**Yunfei Xie[1,*], Ce Zhou[1,*], Lang Gao[1,*], Juncheng Wu[2,*], Xianhang Li[3],**
**Hong-Yu Zhou[4], Sheng Liu[5], Lei Xing[5], James Zou[5], Cihang Xie[3], Yuyin Zhou[3]**
[*]equal technical contribution

[1]Huazhong University of Science and Technology,    [2]Tongji University,
[3]UC Santa Cruz,    [4]Harvard University,    [5]Stanford University

## Abstract

This paper introduces MedTrinity-25M, a comprehensive, large-scale multimodal dataset for medicine, covering over 25 million images across 10 modalities, with multigranular annotations for more than 65 diseases. These enriched annotations encompass both global textual information, such as disease/lesion type, modality, region-specific descriptions, and inter-regional relationships, as well as detailed local annotations for regions of interest (ROIs), including bounding boxes, segmentation masks. Unlike existing approach which is limited by the availability of image-text pairs, we have developed the first automated pipeline that scales up multimodal data by generating multigranular visual and texual annotations (in the form of image-ROI-description triplets) without the need for any paired text descriptions. Specifically, data from over 90 different sources have been collected, preprocessed, and grounded using domain-specific expert models to identify ROIs related to abnormal regions. We then build a comprehensive knowledge base and prompt multimodal large language models to perform retrieval-augmented generation with the identified ROIs as guidance, resulting in multigranular texual descriptions. Compared to existing datasets, MedTrinity-25M provides the most enriched annotations, supporting a comprehensive range of multimodal tasks such as captioning and report generation, as well as vision-centric tasks like classification and segmentation. This dataset can be utilized to support large-scale pre-training of multimodal medical AI models, contributing to the development of future foundation models in the medical domain. The dataset is publicly available at `https://yunfeixie233.github.io/MedTrinity-25M/`.

## 1  Introduction

Large-scale multimodal foundation models [1, 2, 3, 4, 5] have demonstrated remarkable success across various domains due to their ability to understand complex visual patterns in conjunction with natural language. This success has sparked significant interest in applying such models to medical vision-language tasks. Much progress has been made to improve the medical capacity of general domain multimodal foundation models by constructing medical datasets with image-text pairs and fine-tuning general domain models on these datasets [6, 7, 8, 9, 10].

However, current medical datasets have several limitations. Firstly, these datasets lack **multigranular** annotations that reveal the correlation between local and global information within medical images.

Submitted to the 38th Conference on Neural Information Processing Systems (NeurIPS 2024) Track on Datasets and Benchmarks. Do not distribute.

Medical images often contain detailed cues, such as regional abnormal textures or structures, which may indicate specific types of lesions. Therefore, multimodal models need the ability to infer global information, such as disease or lesion type, from local details. The absence of such data limits the models' capacity to comprehensively understand medical images. Moreover, current dataset construction methods heavily rely on medical images paired with reports or captions, which restricts their scalability.

In this paper, we address the above challenges by proposing an automated data construction pipeline using multimodal large language models (MLLMss) without relying on paired text descriptions. To address the lack of comprehensive medical knowledge in general-purpose MLLMs, we leverage domain-specific expert grounding models and retrieval-augmented generation (RAG) to extract relevant medical knowledge. We then prompt MLLMs to generate multigranular visual and textual annotations enriched with this knowledge based on identified regions of interest (ROIs). We utilize this pipeline to transform the collected data, including large-scale unpaired images, into image-ROI-description triplets. These triplets provide multigranular annotations that encompass both global textual information, such as disease/lesion type, modality, and inter-regional relationships, as well as detailed local annotations for ROIs, including bounding boxes, segmentation masks, and region-specific textual descriptions. Using the proposed pipeline, we create a large-scale multimodal multigranular medical dataset containing over 25 million triplets, named **MedTrinity-25M**. To our best knowledge, this is the largest multimodal dataset in medicine to date.

Initially, we assemble a large amount of medical data from over 90 online resources such as TCIA, Kaggle, Zenodo, Synapse, etc. In addition to images with a small amount of high-quality paired manual reports, this assembled data also includes two types of coarse medical data: 1) Image data with segmentation masks, lesion bounding boxes, or only disease types but lacking detailed textual descriptions, and 2) Images paired with coarse captions that describe only global modality or disease information, but lack detailed descriptions of local regions. To generate multigranular annotations from the massive coarse medical data, we first identify ROIs that contain disease or lesion patterns by applying expert grounding models. We then build a comprehensive knowledge base from online corpora (e.g., PubMed) and retrieve image-related medical knowledge. Finally, we prompt MLLMs to integrate medical knowledge with guidance of identified ROIs to generate multigranular textual descriptions.

## 2   Related Work

**Medical Multimodal Foundation Models.**   Due to the effectiveness of multimodal foundation models in understanding visual features, adapting these models to perform medical vision-language tasks has garnered increasing attention in recent years [11, 12, 9, 5]. Several papers attempt to adapt general domain multimodal foundation models with varying architecture to medical domain through end-to-end training on medical datasets. For example, Med-Flamingo [11] enhances the medical capacity of OpenFlamingo-9B [13] by fine-tuning it with 0.8M interleaved and 1.6M paired medical image-text data. While Med-PalM [12] adapts PaLM-E [14] to medical domain using approximately 1M medical data points, demonstrating competitive or surpassing performance compared to state-of-the-art models. Additionally, LLaVA-Med [9] employs end-to-end visual instruction tuning [1] with two stages, achieving remarkable results in medical Visual Question Answering (VQA) tasks. Similarly, Med-Gemini [15] employs a long-form question answering dataset to enhance the multimodal and long-context capabilities of baseline Gemini [16]. Although these models have achieved remarkable performance, they are still limited by the scale of training data. Prior research [17] has shown that scaling up the training data improves the performance of large multimodal foundation models. In this paper, we aim to build a large-scale medical dataset to facilitate the development of more powerful medical multimodal foundation models.

**Multimodal Datasets for medicine.**   The significance of construting comprehensive medical multimodal datasets has garnered considerable attention [9, 18, 19, 7]. Several works attempt to collect images and paired clinical reports prepared by pathology specialist [19, 7, 8], which provide

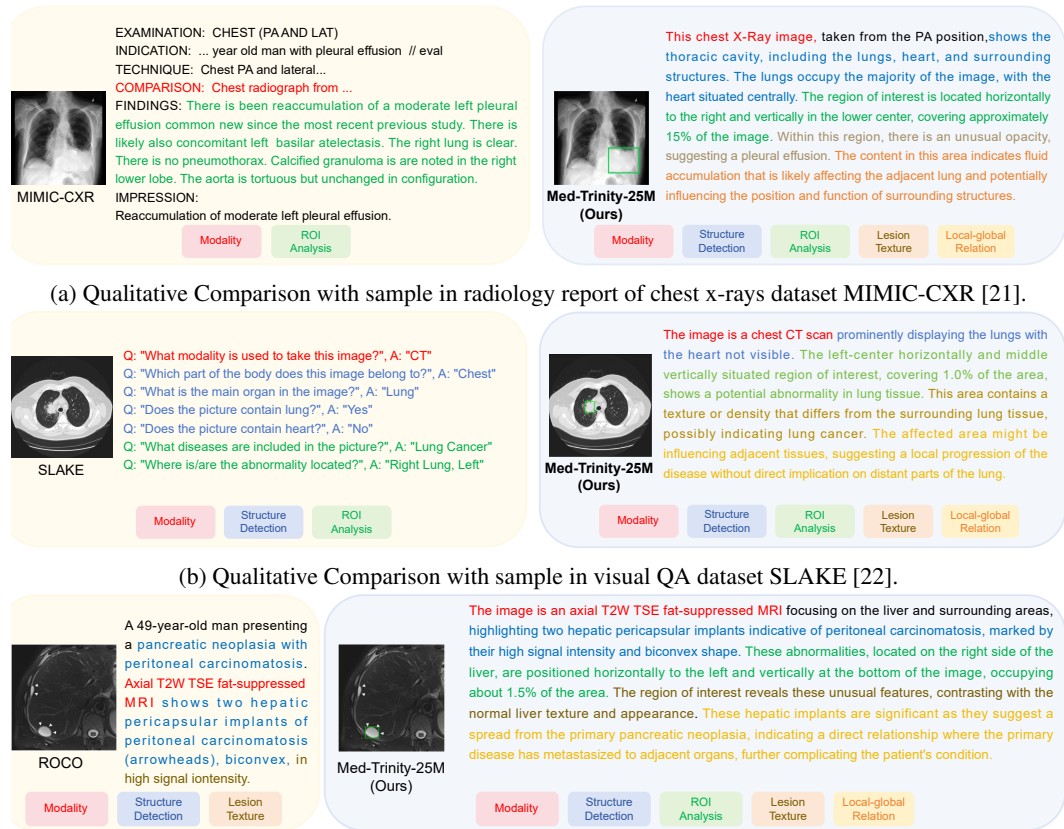

(a) Qualitative Comparison with sample in radiology report of chest x-rays dataset MIMIC-CXR [21].

(b) Qualitative Comparison with sample in visual QA dataset SLAKE [22].

(c) Qualitative Comparison with sample in radiology objects caption dataset ROCO [18].

Figure 1: Qualitative comparison with different types of dataset.

comprehensive descriptions of images, including disease types and corresponding reasoning. For example, MIMIC-CXR[8] comprises 227,835 images for 65,379 patients, containing pathological findings and impressions in reports paired with each images. However, manually constructing such reports is both time-consuming and expensive, thereby limiting the scale of these datasets. PMC-OA [20] aims to expand the dataset scale by extracting a large number of image-caption pairs from medical papers, increasing the number of data samples to 1.65 million. However, the extracted captions are less detailed compared to manual clinical reports, resulting in a lack of multigranular annotations. RadGenome-Chest CT [19] includes more detailed annotations, such as segmentation masks and medical reports generated by MLLMs. Nonetheless, its construction method still relies on paired image-text data, which limits its scalability. Unlike these existing methods, we devise the first automated data construction pipeline to generate multigranular annotations for unpaired images, achieving a comprehensive multigranular dataset with 25 million data samples.

# 3 MedTrinity-25M Dataset

## 3.1 Data Triplet

Our dataset comprises triplets of {image, ROI, description}. Each ROI is associated with an abnormality and is represented by a bounding box or a segmentation mask, specifying the relevant region within the image. For each image, we provide a multigranular textual description, which includes the disease/lesion type, modality, region-specific description, and inter-regional relationships as illustrated in Figure 2.

**Images.** We use the original medical image in the source dataset, we extensively collected medical datasets from the following sources: (1) online resources such as TCIA, Kaggle, Zenodo, Synapse, Hugging Face,Grand Challenge , GitHub, etc. (2) relevant medical dataset research, such as CheX-pert [7] and DeepLesion [23]. These datasets were first categorized into two types: (1) datasets containing local annotations, such as MIMIC-CXR [8] with corresponding radiology reports, and PMC-OA [24] with corresponding captions, where the reports or captions provide analysis of specific local conditions in the images; another example is the 3D image segmentation dataset BraTS2024 [25], which marks the tumor regions in CT scans with masks. (2) datasets containing global annotations: such as image classification datasets ISIC2019 [26] and ISIC2020 [27], whose classification labels reflect the overall pathological condition of tissue sections; another example is the CheXpert [7] dataset, which provides detailed classification of disease types for each chest X-ray. We collect 25,001,668 samples spanning 10 modalities and over 65 diseases. For 3D volumetric images stored in DICOM or NIfTI formats, we converted each 2D slice to PNG format. Additional caption and annotations like masks and bounding boxes from these datasets were utilized to construct ROIs and corresponding textual descriptions as below.

**ROIs.** For each image, ROIs are highlighted using segmentation masks or bounding boxes. These ROIs mostly contain pathological findings such as lesions, inflammation, neoplasms, infections, or other potential abnormalities. In the few cases without abnormalities, the ROIs generally indicate the primary object or organ in the image, as shown in examples in the supplementary material.

**Textual Descriptions.** The textual descriptions for each image are provided with detailed information across various aspects. Unlike the unstructured free-text descriptions found in previous medical report datasets[7, 8, 6] or simple short sentences in visual QA dataset[28, 22] and caption dataset[18, 24], our textual descriptions are multigranular and structured. General attributes related to the image are described first, including the image modality, the specific organ depicted, and the type of disease presented. Subsequently, ROI-related information is provided, including their locations and the abnormal characteristics within them that indicate underlying pathology, such as distinctive color and texture. Additionally, comparisons between the ROIs and surrounding regions are presented to highlight differences in features and the extent of disease progression.

We also demonstrate the multigranular textual descriptions in our dataset with those in other common forms. As illustrated in Figure 1, our textual description is multigranular with more attributes than radiology report of chest x-rays dataset MIMIC-CXR [21], visual QA dataset SLAKE[22] and radiology objects caption dataset ROCO[18].

### 3.2 Data Construction Pipeline

Given a medical image, we aim to generate corresponding multigranular visual and texual annotations by leveraging MLLMs. Specifically, as shown in Figure 2, our pipeline can be decomposed into two stages - **Data Processing** and **Generation of Multigranular Text Description**. In the **Data Processing** stage (Section 3.2.1), we address the lack of domain-specific knowledge in general-purpose MLLMs by leveraging expert grounding models and retrieval-augmented generation (RAG). This stage includes three key steps: 1) **Metadata Integration** to produce coarse captions encapsulating fundamental image information such as modality and disease types; 2) **ROI Locating** to identify regions of abnormalities; and 3) **Medical Knowledge Retrieval** to extract relevant fine-grained medical details. Based on the processed data, we then prompt MLLMs to generate multigranular text descriptions, resulting in the creation of fine-grained captions, as detailed in Section 3.2.2.

### 3.2.1 Data Processing

**Coarse Caption Generation via Metadata Integration.** We aim to generate coarse captions that provide fundamental information for a given image, including modality, organ labels, disease types, and optionally, camera views and equipment information. Instead of extracting features directly from the images, we generate these captions by integrating dataset metadata. We first extract metadata from

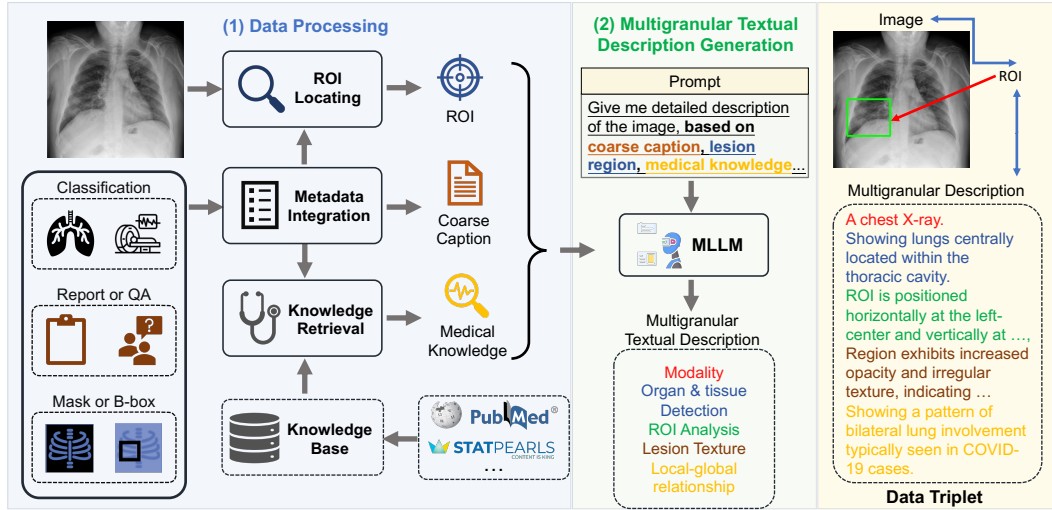

Figure 2: **Data construction pipeline.** 1) Data processing: extracting essential information from collected data, including **metadata integration** to generate coarse caption, **ROI locating**, and **medical knowledge collection**. 2) Multigranular textual description generation: using this information to prompt MLLMs to generate fine-grained captions.

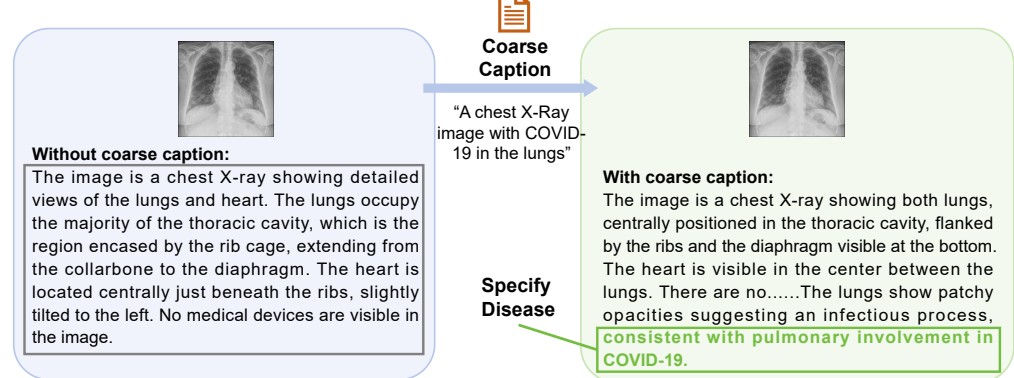

Figure 3: **A qualitative comparison example of generated textual description with and without coarse caption.** Without a coarse caption, MLLMs fails to detect diseases. On the contrary, providing a caption mentioning "COVID-19" allows MLLMs to identify and categorize the disease, facilitating further analysis.

the datasets and then apply a fixed rule to integrate this information into coarse captions. For example, for an image from the QaTa-COV19 dataset[1], we derive metadata from the dataset's accompanying paper or documentation, indicating that it consists of COVID-19 chest X-ray images. Next, we construct coarse captions like "A chest X-ray image with COVID-19 in the lungs" highlighting the modality, organ types, and disease labels. If the image contains additional textual information like radiological findings, this is also integrated to enhance the richness of the caption. The effectiveness of adding coarse captions when generating fine-grained captions is illustrated in Figure 3. In contrast to the scenario without a coarse caption where MLLMs fails to recognize the disease, providing MLLMs with a coarse caption that includes the disease type "COVID-19" enables it to identify and categorize the disease, thereby laying the foundation for further analysis.

**ROI Locating.** We employ various strategies to locate Regions of Interest (ROIs) in images. For datasets that already include localization annotations, such as segmentation masks or bounding boxes, we derive the ROIs from these existing annotations. Specifically, bounding boxes are directly used

---

[1] https://www.kaggle.com/aysendegerli/qatacov19-dataset.

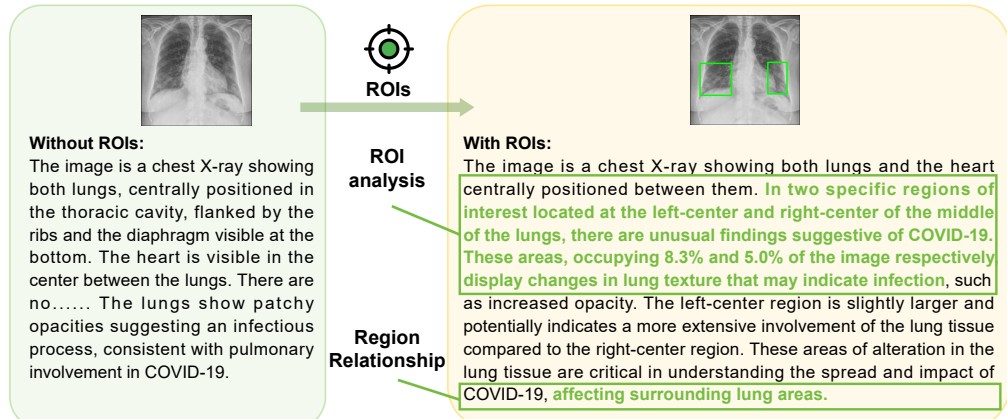

Figure 4: **A qualitative comparison example of generated textual description with and without locating ROIs.** Without ROIs, the caption offers only a brief global analysis; with ROIs, MLLMs conducts detailed local analysis and assesses the impact of lesion ROIs on adjacent normal regions.

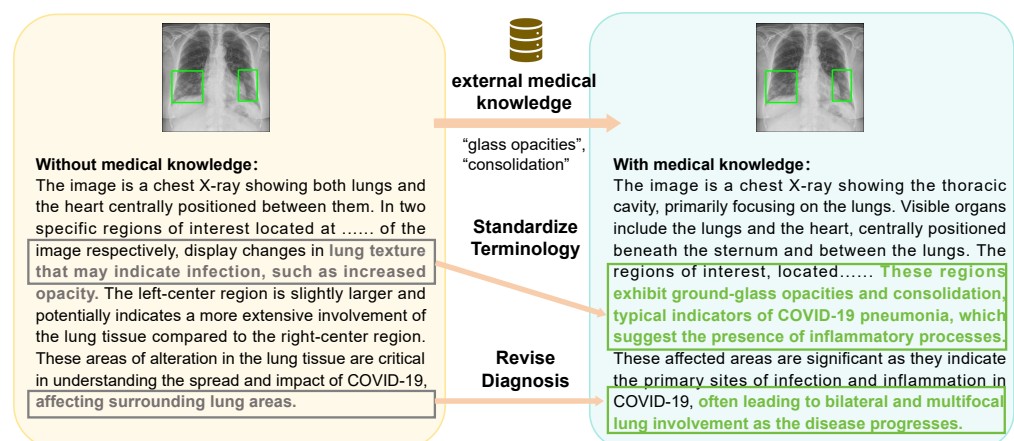

Figure 5: **A qualitative comparison example of generated textual description with and without external medical knowledge.** MLLMs can standardize medical terminology in its expressions and refine its diagnosis based on disease progressions detailed in medical literature.

as the ROIs, while segmentation masks are converted to ROIs by creating the smallest bounding box that covers the mask. When such localization annotations are not available, we apply different pretrained expert models listed in the Appendix to generate ROIs. For text-prompt driven grounding model[29], we use disease and organ information in coarse captions as text prompts to guide the model in segmenting specific parts. Examples of generated ROIs from various modalities with different models are demonstrated in Figure 6.

Without ROIs, the original description is limited to a brief global analysis of the image. However, with ROIs, MLLMs can perform a more detailed local analysis of the ROIs and assess the impact of lesion ROIs on the surrounding normal regions, as demonstrated in Figure 4.

**Medical Knowledge Retrieval.** General-purpose MLLMs often produce content that lacks specialized medical terminology and professional expression. To address this issue, we build a medical knowledge database following the approach in MedRAG [32]. We collect three main corpora: PubMed[2] for biomedical knowledge, StatPearls[3] for clinical decision support, and medical textbooks [33] for domain-specific knowledge. We segment these corpora into short snippets and encode

[2] https://pubmed.ncbi.nlm.nih.gov/
[3] https://www.statpearls.com/

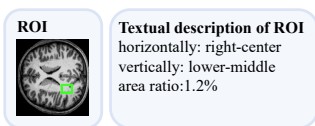

(a) Example of locating ROI via SAT[29].

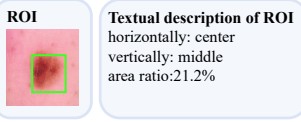

(b) Example of locating ROI via BA-Transformerr [30].

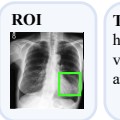

(c) Example of locating ROI via MedRPG [31].

Figure 6: Example of ROIs and their corresponding textual descriptions.

**Knowledge 1:**
**Title: Mobile chest X-ray manifestations of 54 deceased patients with coronavirus disease 2019: Retrospective study.**
Content: ...... We found that 50 (93%) patients with **lesions occurred in the bilateral lung**, 4 (7%) patients occurred in the right lung, 54 (100%) patients were **multifocal involvement**. The number of lung fields involved was 42 (78%) patients in 6 fields, 3 (6%) patients in 5 lung fields, 4 (7%) patients in 4 lung fields, and 5 (9%) patients in 3 lung fields. Fifty-three (98%) patients had **patchy opacities**, 3 (6%) patients had round or **oval solid nodules**, 9 (17%) patients had fibrous stripes, 13 (24%) patients had **pleural effusion**, 8 (15%) patients had **pleural thickening**, 6 (11%) patients had **pneumothorax**, 3 (6%) patients had **subcutaneous emphysema**. Among the 24 patients who had serial mobile chest X-rays, 16 (67%) patients had the progression of the lesions, 8 (33%) patients had no significant change of the lesions, and there was no case of reduction of the lesions.The mobile chest X-ray manifestations of deceased patients with COVID-19 were **mostly bilateral lung, multifocal involvement, and extensive lung field, and pleural effusion, pleural thickening, and pneumothorax probably could be observed.** The serial mobile chest X-ray showed that the chest lesions were progressive with a high probability.
.......

Figure 7: **An example of the Top-8 retrieval results.** By leveraging COVID-19-related medical knowledge, MLLMs can standardize medical terminology and enhance diagnoses according to the disease progressions described in medical literature.

them into high-dimensional vectors using the text encoder from Med-CPT [34]. These vectors are then indexed into a specialized vector knowledge base using Faiss[35], optimized for efficient retrieval.

For a given image, we retrieve relevant medical knowledge by using its coarse caption, which is generated through metadata integration. Specifically, we encode the coarse captions, including disease and organ classifications, into vectors using the Med-CPT text encoder. We then perform a vector similarity search in the medical vector database, retrieving the top eight medical knowledge snippets that semantically match the query. These snippets provide the external medical knowledge paired with the image. A qualitative example demonstrating the effectiveness of incorporating external medical knowledge is shown in Figure 7. With access to COVID-19-related medical knowledge, MLLMs can standardize medical terminology and refine diagnoses based on the disease progressions outlined in medical literature.

### 3.2.2 Generation of Multigranular Text Description

After data processing, a comprehensive prompt is utilized to guide the MLLMs in generating multi-granular descriptions. The prompt template consists of a three-level hierarchical framework with questions to instruct MLLMs: (1) a global description that captures all details of the image; (2) a local-focused analysis of specific ROIs that potentially are unusual; and (3) a local-global examination of the interaction between local and global attributes to understand the impact of local abnormalities on the entire organ. Detailed prompt template is presented in supplementary materials.

To ensure that the MLLMs are guided by relevant medical information not inherently present in their training data, we incorporate the processed data (coarse captions, ROIs, and retrieved medical knowledge) into the prompts. Specifically, for global information, coarse captions are directly integrated into the prompt. For local information, ROIs on images are converted into textual descriptions based on their coordinates and area ratio within the images. Examples of these textual descriptions are shown in Figure 6, using terms such as "left-center" and "area ratio: 1.2%."

To refine terminology and diagnosis within ROIs, relevant medical knowledge about specific diseases is incorporated into the prompt. Instead of merely inserting this knowledge, we instruct MLLMs to identify and align the relevant knowledge to ROIs that require analysis.

**Choice of MLLMs** We first prompt GPT-4V with the provided medical coarse captions, ROIs, and medical knowledge to generate a subset of 200,000 samples, maintaining a similar modality and organ distribution to our full 25 million dataset. The goal of curating this subset is to calibrate a medical knowledge-guided MLLM to adhere to the formatting instructions specified for our text.

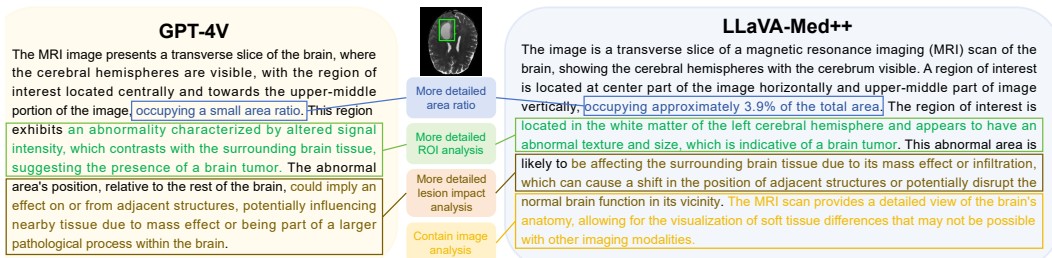

Figure 8: **Qualitative Comparison with sample generated by GPT-4V** Compared to GPT-4V, our model generate more detailed caption.

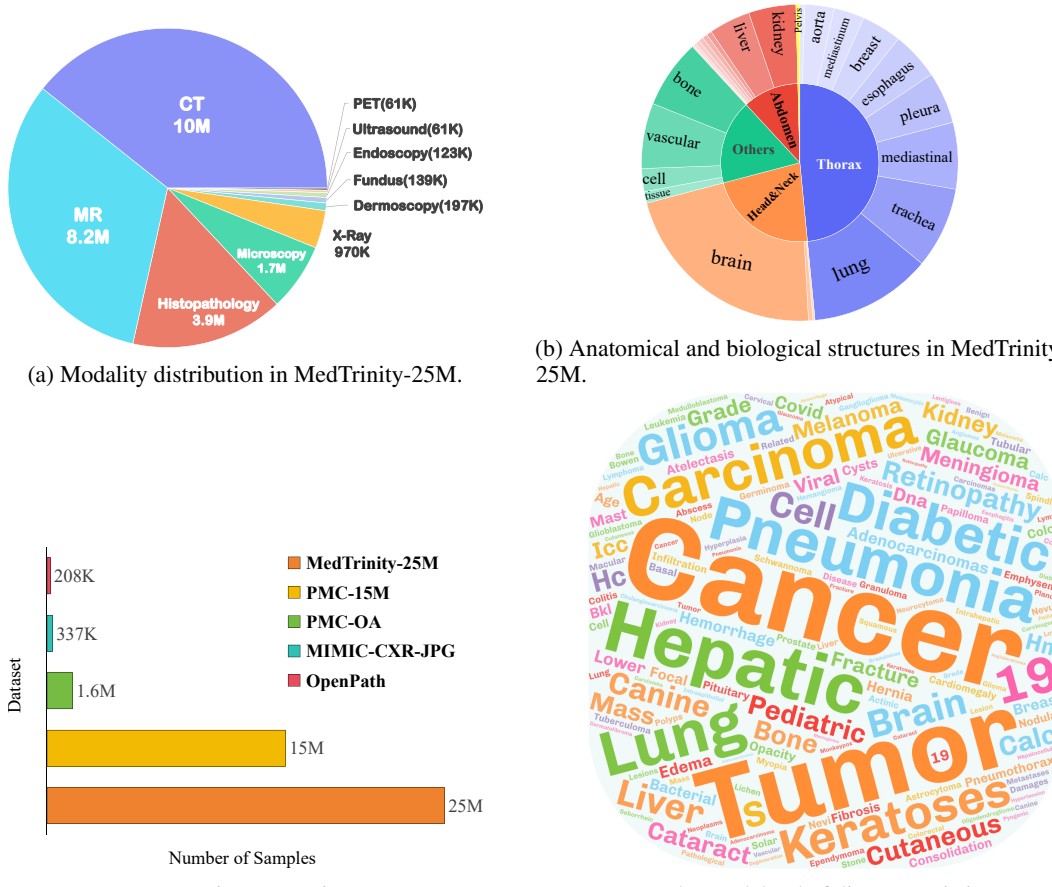

(a) Modality distribution in MedTrinity-25M.

(b) Anatomical and biological structures in MedTrinity-25M.

(c) Data size comparison.

(d) Wordcloud of disease statistic.

Figure 9: Statistical overview of MedTrinity-25M.

Subsequently, we employ our model, LLaVA-Med++, which is based on LLAVA-Med [9], the state-of-the-art medical MLLM. To further improve this model, we leverage the latest LLaMA3[36] to enhance its linguistic capabilities, and incorporate multi-scale feature extraction [37] to improve its vision capabilities. LLaVA-Med++ undergoes continuous training on medical multimodal data and is fine-tuned using our multigranular annotations, resulting in a specialized medical model.

After fine-tuning, we then use this specialized model to generate the multigranular text descriptions on our entire dataset, resulting in 25 million image-ROI-description triplets. The fine-tuning process leverages the advanced language organization capabilities of GPT-4V, providing an effective template for fine-grained captions, which our model uses to learn the formatting of fine-grained captions. As a result, our model generates more detailed descriptions compared to GPT-4V, as illustrated in Figure 8. We also show a detailed quantitative comparison in the supplementary material.

| Dataset | Modality | Lesion Type | Lesion BBox/Mask | Color Texture Description | Region Relationship |
|---|---|---|---|---|---|
| MedMNIST [39] | ✗ | ✓ | ✗ | ✗ | ✗ |
| DeepLesion [40] | ✓ | ✗ | ✓ | ✗ | ✗ |
| BraTS 2024 [41] | ✓ | ✗ | ✓ | ✗ | ✗ |
| MIMIC-CXR [21] | ✓ | ✓ | ✓ | ✓ | ✗ |
| Quilt-1M [10] | ✓ | ✓ | ✗ | ✓ | ✓ |
| VQA-RAD [42] | ✓ | ✓ | ✗ | ✓ | ✗ |
| CRC100K [43] | ✓ | ✓ | ✗ | ✗ | ✗ |
| SA-Med2D-20M [44] | ✓ | ✓ | ✓ | ✗ | ✗ |
| **MedTrinity-25M(Ours)** | ✓ | ✓ | ✓ | ✓ | ✓ |

Table 1: Comparison of dataset types based on provided attributes of annotations.

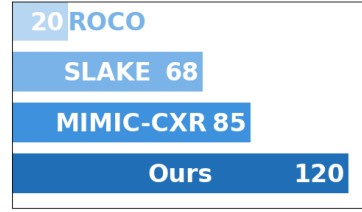

Figure 10: **Comparison of the average word count of text descriptions.**

## 4   Dataset Analysis

**Diversity**   Our dataset encompasses a wide range of 10 imaging modalties, with more than 65 diseases across various anatomical structures in human. The distribution of Anatomical and biological structures in MedTrinity-25M is shown in Figure 9b. Meanwhile, the number of samples in the dataset for each modality are shown in Figure 9a, spanning from common ones with over 1 million samples each (CT, MRI, X-ray) to rare modalities(ultrasound, dermoscopy) with at least more than 100,000 samples, demonstrating a much more balanced distribution compared to other large-scale dataset like SA-Med2D-20M[38], which only contain thousands of ultrasound and dermoscopy samples.

**Scale**   Figure 9c shows the amount of our dataset, which is significantly larger than previous datasets. To the best of our knowledge, this is the largest open-source, multi-modal multigranular medical dataset to date.

**Diseases**   The datasets involved in constructing MedTrinity-25M primarily focus on disease diagnosis and medical discovery. In MedTrinity-25M, diseases are given in the free-form text. The same disease may be referred to using different terms, allowing for elaborate identification and analysis. Figure 9d illustrates the frequently used words related to diseases in our dataset.

**Richness**   We provide both quantitative analysis and qualitative examples to show the richness of our generated multigranular compare to other medical dataset. Qualitative examples are shown in Figure 1, our textual description is multigranular with more attributes than radiology report of chest x-rays dataset MIMIC-CXR [21], visual QA dataset SLAKE[22] and radiology objects caption dataset ROCO[18]. To demonstrate the multi-granularity of our data, we compared the average word count of text descriptions in our dataset, MedTrinity-25M, with those in other medical datasets, as illustrated in Figure 10. The word count in our dataset is significantly higher, indicating greater richness.

**Alignment with human**   We leverage GPT-4 to quantify the alignment of generated text descriptions compared to clinical reports from pathologist, which is set as the ground-truth. Specifically, we utilize GPT-4 to score the helpfulness, relevance, accuracy, and level of details of the our generated text descriptions based on clinical reports, and give an overall score on a scale of 1 to 10, where a higher score indicates better overall performance. Additionally, GPT-4 is required to provide a comprehensive explanation for the evaluation score. Detailed experiment results are presented in supplementary materials.

## 5   Conclusion

This paper introduces MedTrinity-25M, a large-scale multimodal medical dataset comprising over 25 million image-ROI-description triplets sourced from more than 90 online resources, spanning 10 modalities and covering over 65 diseases. Unlike existing dataset construction methods that rely on image-text pairs, we have developed the first automated pipeline to scale up multimodal data by generating multigranular visual and textual annotations from unpaired image inputs, leveraging expert grounding models, retrieval-augmented generation techniques, and advanced MLLMs. MedTrinity-25M's enriched annotations have the potential to support a wide range of multimodal tasks, such as captioning, report generation, classification, and segmentation, as well as facilitate the large-scale pre-training of multimodal medical AI models.

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
