# MedTrinity-25M: A Large-scale Multimodal Dataset with Multigranular Annotations for Medicine

**Yunfei Xie**[1,*], **Ce Zhou**[1,*], **Lang Gao**[1,*], **Juncheng Wu**[2,*], **Xianhang Li**[3],
**Hong-Yu Zhou**[4], **Sheng Liu**[5], **Lei Xing**[5], **James Zou**[5], **Cihang Xie**[3], **Yuyin Zhou**[3]
[*]equal technical contribution

[1]Huazhong University of Science and Technology,      [2]Tongji University,
[3]UC Santa Cruz,      [4]Harvard University,      [5]Stanford University

## 1 Supplementary material

## 2 A  Data Source

Table 1: Data sources for MedTrinity-25M from various medical image datasets, detailing their modalities, biological structures, quantities, and annotations.

| Dataset Name | Modality | Biological Structures | Quantity | Text | Disease Type | BBox | Mask |
|---|---|---|---|---|---|---|---|
| BCNB [1] | Histopathology | breast | 76579 | ✗ | ✓ | ✗ | ✗ |
| BHX [2] | CT | brain | 1831797 | ✗ | ✗ | ✓ | ✗ |
| BKAI-IGH [3] | Endoscopy | colon | 1000 | ✗ | ✗ | ✗ | ✓ |
| Blood Cell [4] | Microscopy | cell | 12500 | ✗ | ✓ | ✗ | ✗ |
| Bone Fracture [5] | X-Ray | bone | 4148 | ✗ | ✗ | ✗ | ✓ |
| Brain MRI-seg [6] | MR | brain | 7860 | ✗ | ✗ | ✗ | ✓ |
| Brain Tumor-seg [7] | MR | brain | 3064 | ✗ | ✗ | ✗ | ✓ |
| Brain-Tumor-Detection [8] | MR | brain | 9900 | ✗ | ✗ | ✗ | ✓ |
| BRATS2024 [9] | MR | brain | 1486406 | ✗ | ✗ | ✗ | ✓ |
| Breast Pathology [10] [11] | Histopathology | breast | 555048 | ✗ | ✓ | ✗ | ✗ |
| Breast Ultrasound [12] | Ultrasound | breast | 514 | ✗ | ✓ | ✗ | ✗ |
| breastcancer [13] | Histopathology | breast | 20000 | ✗ | ✗ | ✗ | ✓ |
| BREAST-LESIONS-USG [14] | Ultrasound | breast | 253 | ✗ | ✗ | ✗ | ✓ |
| BTCV-cervix [15] | CT | cervix | 11695 | ✗ | ✗ | ✗ | ✓ |
| BUS-BRA [16] | Ultrasound | breast | 1876 | ✗ | ✗ | ✗ | ✓ |
| BUSI-with-GT [17] | Ultrasound | breast | 648 | ✗ | ✗ | ✗ | ✓ |
| Capstone v3 [18] | Dermoscopy | skin | 12532 | ✗ | ✓ | ✗ | ✗ |
| CBIS-DDSM-cls [19, 20, 21] | X-Ray | breast | 10239 | ✗ | ✓ | ✗ | ✗ |
| CBIS-DDSM-seg [22] | X-Ray | breast | 6206 | ✗ | ✗ | ✗ | ✓ |
| CheXpert [23] | X-Ray | lung | 223648 | ✗ | ✓ | ✗ | ✗ |
| CholecSeg8k [24] | Endoscopy | colon | 32300 | ✗ | ✗ | ✗ | ✓ |
| COVID-19 CXR [25] [26] | X-Ray | lung | 10956 | ✗ | ✓ | ✗ | ✗ |
| QU-Ex [27, 28, 29, 30] | X-Ray | lung | 26990 | ✗ | ✗ | ✗ | ✓ |
| COVIDx [31] | X-Ray | lung | 61441 | ✗ | ✓ | ✗ | ✗ |
| CPD-seg [32] | Histopathology | skin | 202 | ✗ | ✗ | ✗ | ✓ |
| CR-AI4SKIN [33] | Histopathology | skin | 53122 | ✗ | ✓ | ✗ | ✗ |
| CRC100K [34] | Histopathology | colon | 100000 | ✗ | ✓ | ✗ | ✗ |
| Crystal Clean [35] | MR | brain | 18606 | ✗ | ✓ | ✗ | ✗ |
| CT2USforKidneySeg [36] | Ultrasound | breast | 4586 | ✗ | ✗ | ✗ | ✓ |

Submitted to the 38th Conference on Neural Information Processing Systems (NeurIPS 2024) Track on Datasets and Benchmarks. Do not distribute.

Table 1 : Continued from previous page

| Dataset Name | Modality | Biological Structures | Quantity | Text | Disease Type | BBox | Mask |
|---|---|---|---|---|---|---|---|
| CT-RATE [37] | CT | lung, liver, mediastinum, kidney, heart,etc. | 4624426 | ✓ | ✗ | ✗ | ✗ |
| CXR-pneumothorax [38] | X-Ray | lung | 2492 | ✗ | ✗ | ✗ | ✓ |
| CytoImageNet [39] | Microscopy | cell | 890737 | ✗ | ✓ | ✗ | ✗ |
| DeepLesion [40] | CT | bone, abdomen, mediastinum, liver, lung, kidney, soft tissue, pelvis | 2870411 | ✗ | ✗ | ✓ | ✗ |
| Diabetic Retinopathy [41] | Fundus | eye | 18624 | ✗ | ✓ | ✗ | ✗ |
| Figshare Brain Tumor [42] | MR | brain | 3065 | ✗ | ✗ | ✗ | ✓ |
| HAM10000 [43, 44] | Dermoscopy | skin | 10015 | ✗ | ✓ | ✗ | ✗ |
| Histology [45] | Histopathology | lung | 1608060 | ✗ | ✓ | ✗ | ✗ |
| ihc4bc [46] | Microscopy | cell | 184949 | ✗ | ✓ | ✗ | ✗ |
| isic2019 [47] [48] [44] | Dermoscopy | skin | 25332 | ✗ | ✓ | ✗ | ✗ |
| isic2020 [49] | Dermoscopy | skin | 6838 | ✗ | ✓ | ✗ | ✗ |
| ISPY1 [50] | MR | breast | 386336 | ✗ | ✓ | ✗ | ✗ |
| ISPY2 [51] [52] | CT | breast | 330454 | ✗ | ✗ | ✗ | ✓ |
| Kidney Stone [53] | CT | kidney | 1300 | ✗ | ✗ | ✓ | ✗ |
| KiPA22 [54, 55, 56, 57] | CT | kidney | 29458 | ✗ | ✗ | ✗ | ✓ |
| KiTS23-remain [58] | CT | kidney | 17628 | ✗ | ✗ | ✗ | ✓ |
| Kvasir-seg [59] | Endoscopy | colon | 1000 | ✗ | ✗ | ✗ | ✓ |
| LC25000-colon [60] | Histopathology | colon | 5000 | ✗ | ✓ | ✗ | ✗ |
| LC25000-lung [60] | Histopathology | lung | 10000 | ✗ | ✓ | ✗ | ✗ |
| Leukemia-cls [61] | Microscopy | cell | 15135 | ✗ | ✓ | ✗ | ✗ |
| LiTS2017 [62] | CT | liver | 129900 | ✗ | ✗ | ✗ | ✓ |
| LLaVA-Med [63] | CT, MR, Endoscopy, X-Ray, Ultrasound, Histopathology, Dermoscopy, Microscopy, Fundus, PET | cell, rib, tissue, face, brain, vascular, liver, bone, lymph, etc. | 342214 | ✓ | ✗ | ✗ | ✗ |
| LLD–MMRI2023 [64] | MR | liver | 30956 | ✗ | ✗ | ✗ | ✓ |
| LNQ [65] | CT | lung | 17211 | ✗ | ✗ | ✗ | ✓ |
| MIDOG22 [66] | Histopathology | cell | 20554 | ✗ | ✓ | ✗ | ✗ |
| MIMIC-CXR-JPG [67] | X-Ray | lung | 148624 | ✗ | ✓ | ✗ | ✗ |
| Nerve-Ultrasound-Seg [68] | Ultrasound | breast | 2324 | ✗ | ✗ | ✗ | ✓ |
| NIH CXR-cls [69, 70, 71] | X-Ray | lung | 50879 | ✗ | ✓ | ✗ | ✗ |
| NIH CXR-od | X-Ray | lung | 984 | ✗ | ✗ | ✓ | ✗ |
| padchest [72] | CT | lung | 160861 | ✗ | ✓ | ✗ | ✗ |
| PatchGastricADC22 [73] | Histopathology | gastral | 262000 | ✗ | ✓ | ✗ | ✗ |
| PMC-OA [74] | CT,MR, Endoscopy, X-Ray, Ultrasound, Histopathology, Dermoscopy, Microscopy, Fundus, PET | cell, tissue, vascular, brain, bone, liver, lymph, eye, epithelium,etc. | 1426450 | ✓ | ✗ | ✗ | ✗ |

Table 1 : Continued from previous page

| Dataset Name | Modality | Biological Structures | Quantity | Text | Disease Type | BBox | Mask |
|---|---|---|---|---|---|---|---|
| PMC-VQA [75] | CT, MR, Endoscopy, X-Ray, Ultrasound, Histopathology, Dermoscopy, Microscopy, Fundus, PET | cell, brain, tissue, artery, bone, face, rib, vascular, liver, eye,etc. | 203798 | ✓ | ✗ | ✗ | ✗ |
| QAMEBI [76] [77] [78] | Ultrasound | breast | 232 | ✗ | ✗ | ✗ | ✓ |
| QATA-cls [79, 80, 81, 82, 83] | X-Ray | lung | 17855 | ✗ | ✓ | ✗ | ✗ |
| QATA-seg | X-Ray | lung | 13862 | ✗ | ✗ | ✗ | ✓ |
| Quilt-1M [84] | Histopathology | tissue | 1017712 | ✗ | ✓ | ✗ | ✗ |
| Retinal OCT Images [85] | Fundus | eye | 57919 | ✗ | ✓ | ✗ | ✗ |
| ROCO [86] | CT, MR, Endoscopy, X-Ray, Ultrasound, Histopathology, Dermoscopy, Microscopy, Fundus,PET | artery, bone, tissue, vascular, brain, renal, liver, pelvis, bladder,etc. | 58503 | ✓ | ✗ | ✗ | ✗ |
| RSNA-Pneumonia [87] | X-Ray | lung | 21376 | ✗ | ✗ | ✓ | ✗ |
| SA-SAM-Med2d [88] | X-Ray, PET, CT, MR, Endoscopy, dermoscopy | brain, kidney, liver, lung, pancreas, pulmonary, hepatic, skin,etc. | 5243382 | ✗ | ✗ | ✗ | ✓ |
| SICAPv2 [89] | Histopathology | prostate | 18784 | ✗ | ✓ | ✗ | ✓ |
| SIIM_Pneumothorax [90] | X-Ray | lung | 24178 | ✗ | ✗ | ✗ | ✓ |
| skin cancer [91] [92] [93] | Dermoscopy | skin | 206 | ✗ | ✗ | ✗ | ✓ |
| SyntheticCXR [94] | X-Ray | lung | 104801 | ✗ | ✓ | ✗ | ✗ |
| WSSS4LUAD_cls [95] | Histopathology | lung | 10092 | ✗ | ✓ | ✗ | ✗ |
| WSSS4LUAD_seg [95] | Histopathology | lung | 369 | ✗ | ✗ | ✗ | ✓ |
| **Total** | | | **25001668** | | | | |

## B Evaluation of Alignment to Human Annotations

To evaluate the validity and quality of the generated multigranular annotations, we compared them with their original human annotations to assess the degree of alignment (for samples with human annotations).

Since the generated multigranular annotations contains structured descriptions that may significantly differ from free-text radiology reports and question-answering pairs, we leveraged GPT-4V's vision and language understanding capabilities. Rather than focusing on the exact alignment of sentence structure or organization, GPT-4V assessed the alignment based on the accuracy of medical facts and diagnoses. Specifically, the structure of the generated multigranular annotations consists of five key attributes that characterize a medical image: modality, structure detection, ROI analysis, lesion texture, and local-global relation. To evaluate the generated data, we had GPT-4V perform a detailed comparison with human annotations based on these five attributes. Each attribute was scored on a scale from 0 to 2 points, with a maximum possible total score of 10 points.

We conducted an alignment study on SLAKE [96] and MIMIC-CXR [97], randomly selecting 50 samples to compare with multigranular annotations for evaluating alignment scores against human annotations. As shown in Table 2, the alignment scores were 8.2 and 8.9 for SLAKE and MIMIC-CXR, respectively. The criteria of modality, structure detection, and ROI analysis nearly achieved perfect scores, demonstrating the validity and

Table 2: Comparison of alignment scores between our generated multigranular annotations and human annotations.

(a) Alignment Scores on SLAKE

| Score | SLAKE | | | | | |
|---|---|---|---|---|---|---|
| | Overall | Modality | Structure Detection | ROI Analysis | Lesion Texture | Local-Global Relation |
| Ours | 8.2/10.0 | 2.0/2.0 | 1.7/2.0 | 1.8/2.0 | 1.6/2.0 | 1.1/2.0 |

(b) Alignment Scores on MIMIC-CXR

| Score | MIMIC-CXR | | | | | |
|---|---|---|---|---|---|---|
| | Overall | Modality | Structure Detection | ROI Analysis | Lesion Texture | Local-Global Relation |
| Ours | 8.9/10.0 | 2.0/2.0 | 1.9/2.0 | 1.8/2.0 | 1.6/2.0 | 1.6/2.0 |

Figure 1: **An example of a perfect score result evaluated by GPT-4V.** GPT-4V assesses five criteria, each fully aligned with human annotations, resulting in perfect scores.

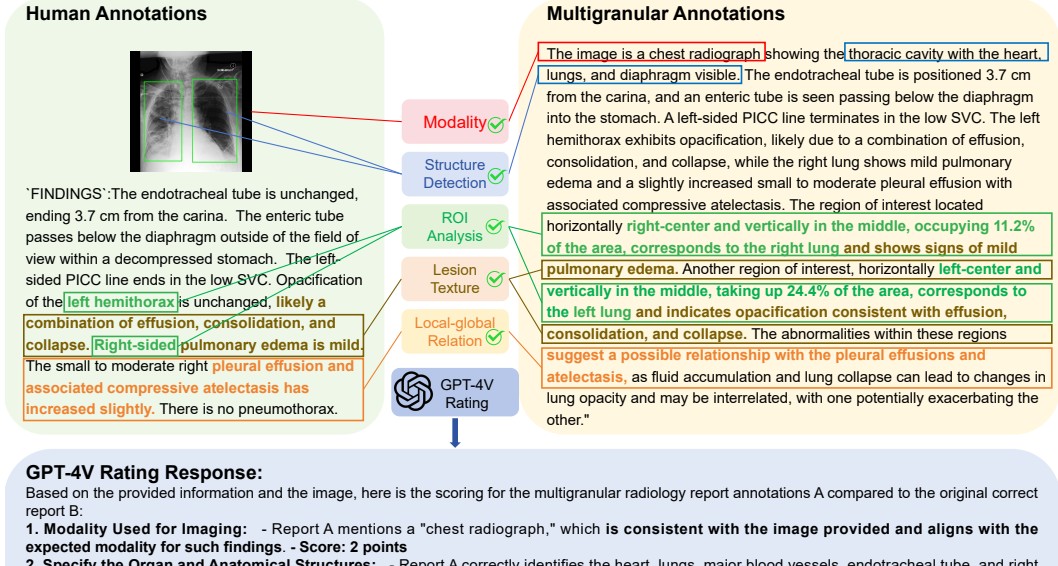

accuracy of the generated data compared to human annotations. An example of perfect alignment score results evaluated by GPT-4V is shown in Figure 1. In these examples, GPT-4V fully aligned with human annotations across all five criteria, resulting in perfect alignment scores.

The prompt used to query GPT-4V for evaluating the alignment score is shown in Figure 2.

Figure 2: Prompt used to evaluate the alignment of generated multigranular annotations.

## Prompting MLLMs to evaluate the alignment of generated multi-granular annotations with human annotations

Let's think it step by step. Evaluate the multigranular radiology report annotations (Report A) compared to the radiology report B step by step. Both reports are based on the same image. Follow these guidelines to ensure accurate assessment:

**Note:** If neither the original question nor radiology report B mentions any abnormalities or diseases, such as "the lungs are clear without confluent consolidation or effusion" or "no pneumothorax is seen", skip the evaluation and return "None."

### Basic Rating Rules:

1. Evaluate each attribute in Report A against radiology report B and verify the information by analyzing the image. Do not deduct points without image analysis.
2. Judge correctness based on the accuracy of medical facts and diagnoses, not on the exact alignment of sentence structure or organization.
3. If radiology report B does not mention any abnormalities or diseases, skip the evaluation and return "None," such as "the lungs are clear without confluent consolidation or effusion" or "no pneumothorax is seen".
4. Each of the 5 attributes should be judged independently. Errors in one attribute should not affect the scoring of other attributes.

### Attributes and Corresponding Rating Rules:

1. **Modality Used for Imaging:**
- **Rating Rule:** Compare with radiology report B. Different names for the same modality (e.g., "chest X-ray" and "CXR") are acceptable.
2. **Specify the Organ and Anatomical Structures:**
- **Rating Rule:** Check if the organs and anatomical structures in Report A match those in radiology report B or appear in the image.
   - Mentioned in both: 2 points
   - Mentioned in one: 1 point
   - Not mentioned in either: 0 points
   - Do not deduct points without image analysis.
3. **Locations of ROI (Regions of Interest):**
- **Rating Rule:** Compare the "horizontal" and "vertical" positions, and the "area ratio" of ROIs with radiology report B. A 5% error in the area ratio is acceptable. If Report A includes at least one ROI from radiology report B, no points are deducted, even if all ROIs are not covered.
4. **Analysis of Abnormal Characteristics:**
- **Rating Rule:** Characteristics indicating pathology should match those in radiology report B or appear in the image.
   - Mentioned in both: 2 points
   - Mentioned in one: 1 point
   - Not mentioned in either: 0 points
   - Do not deduct points without image analysis.
5. **Comparison of Lesions and Surrounding Regions:**
- **Rating Rule:** Differences in features and disease progression should match those in radiology report B or appear in the image.
   - Mentioned in both: 2 points
   - Mentioned in one: 1 point
   - Not mentioned in either: 0 points
   - Do not deduct points without image analysis.

**Note:** Return the scores in a list. For example, if attributes 4 and 5 get deducted 1 point each, while others score 2 points each, return [2, 2, 2, 1, 1]. Provide a short reason (within 80 words) for each point deduction.

Table 3: **Quantitative results of pre-training using our multigranular annotations.** The symbol ✓under 'w/ MedTrinity-25M' indicates that the model has been pre-trained on the MedTrinity-25M dataset prior to training on the target dataset, while ✗ indicates no such pre-training. Multigranular annotations are reformatted to fit with the question and answer format.

| Method | w/ MedTrinity-25M | VQA-RAD | | | SLAKE | | |
| --- | --- | --- | --- | --- | --- | --- | --- |
| | | Open | Close | Overall | Open | Close | Overall |
| GPT-4V [98] | ✗ | 39.5 | 78.9 | 59.2 | 33.6 | 43.6 | 38.6 |
| LLaVA-Med | ✗ | 55.5 | 66.5 | 61.0 | 70.6 | 54.5 | 62.6 |
| LLaVA-Med++ | ✗ | 64.6 | 77.0 | 70.8 | 79.3 | 84.0 | 81.7 |
| LLaVA-Med++ | ✓ | 70.3 | 79.4 | 74.9 | 80.4 | 84.3 | 82.4 |

Figure 3: Examples of ROIs for normal regions.

(a) A no infection sample from MIMIC-CXR. The ROIs highlight the left and right lungs.

(b) A healthy sample from SLAKE. The ROI points out the liver.

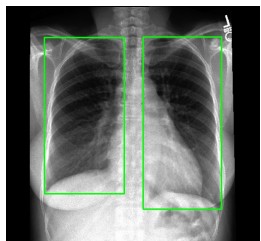

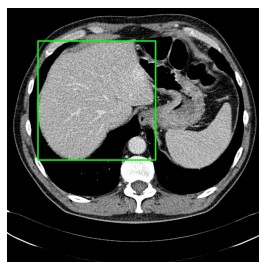

## C  Quantitative Comparison of LLaVA-Med++ with GPT-4V

As detailed in Section 3.2.2 of the main paper, we developed an enhanced version of LLaVA-Med [63], called LLaVA-Med++. This enhancement leverages the latest LLaMA3 [99] to boost linguistic capabilities and incorporates multi-scale feature extraction [100] to improve vision capabilities.

To justify the selection of our specialized medical model, LLaVA-Med++, over GPT-4V for generating textual descriptions, we conducted a quantitative comparison of the outputs generated by both models. We assessed the level of detail by comparing the average word count of text descriptions generated for the same sample. As shown in Figure 4, LLaVA-Med++, after task-specific fine-tuning, outperformed GPT-4V by 3.6% in word count, indicating that the descriptions generated by LLaVA-Med++ are more detailed. Based on these findings, we selected LLaVA-Med++ to generate multigranular annotations for our entire MedTrinity-25M.

## D  MedTrinity-25M Enhances Medical Visual Question Answering (VQA)

To further demonstrate the validity of our dataset, we compare the performance of LLaVA-Med++ with and without training on our dataset. We select Visual Question Answering (VQA) as the evaluation task, which requires models to learn detailed visual and language representations. We assessed the performance of our model on two biomedical VQA datasets: VQA-RAD [101] and SLAKE [96].

We initially pretrained LLaVA-Med++ using the LLaVA-Med [63] methodology as our baseline. Then, we augmented our training data with MedTrinity-25M to develop our final model. Finally, we fine-tuned the model on the VQA datasets for three epochs and evaluated its performance, as shown in Table 3. Comparing results from the same architecture with and without MedTrinity-25M pretraining, it is evident that pretraining with MedTrinity-25M significantly enhances performance.

Specifically, LLaVA-Med++ boosts performance by approximately 4.1% on VQA-RAD and 0.7% on SLAKE compared to training the model from scratch without pretraining on MedTrinity-25M. This improvement demonstrates the effectiveness of pretraining on MedTrinity-25M for downstream multimodal medical tasks such as VQA.

## E  Examples of ROIs for Normal Regions

As detailed in Section 3.1 of the main paper, the regions of interest (ROIs) identified using expert grounding models predominantly contain pathological findings such as lesions, inflammation, neoplasms, infections, or

Figure 4: Qualitative comparison of the relative average word count of samples generated by LLaVA-Med++ and GPT-4V.

| | |
|---|---|
| **LLaVA-Med++(Ours)** | 103.6% |
| **GPT-4V** | 100.0% |

Table 4: List of expert models used to generate ROIs for different datasets.

| ID | Dataset Name | Model |
|----|--------------|-------|
| 1 | Histology | |
| 2 | Quilt-1M | |
| 3 | CytoImageNet | |
| 4 | PatchGastricADC22 | |
| 5 | hc4bc | |
| 6 | CRC100K | |
| 7 | BCNB | |
| 8 | MIDOG22 | Cellpose [102] |
| 9 | Leukemia-cls | |
| 10 | Blood Cell | |
| 11 | WSSS4LUAD_cls | |
| 12 | LC25000-colon | |
| 13 | LC25000-lung | |
| 14 | CR-AI4SKIN | |
| 15 | chexpert | |
| 16 | SyntheticCXR | |
| 17 | ROCO | |
| 18 | NIH CXR-cls | SAT [103] |
| 19 | Crystal Clean | |
| 20 | QATA-cls | |
| 21 | CBIS-DDSM-cls | |
| 22 | PMC-OA | |
| 23 | ISPY1 | |
| 24 | LLaVA-Med | SAM-Med-2D [104] |
| 25 | PMC-VQA | |
| 26 | ISIC2019 | |
| 27 | ISIC2020 | |
| 28 | Capstone v3 | BA-Transformer [105] |
| 29 | HAM10000 | |
| 30 | padchest | |
| 31 | MIMIC-CXR-JPG | CheXmask [106] [107] |
| 32 | COVIDx | |
| 33 | COVID-19 CXR | MedRPG [108] |
| 34 | Diabetic Retinopathy | retina-features [1] |

other potential abnormalities. In the few instances where no abnormalities are present, the ROIs typically highlight the primary object or organ in the image. Examples of ROIs without abnormalities are shown in Figure 3.

## F   List of Expert models to locate ROIs

As detailed in Section 3.2.1 of the main paper, for datasets lacking localization information such as segmentation masks and bounding boxes, we employ various pretrained expert models to identify the ROIs. The specific expert models used for each dataset are listed in Table 4.

# G    Prompt Template for Generation of Multigranular Text Description

To generate multigranular textual descriptions, we design a multi-task prompting approach, breaking down this task into several smaller descriptive tasks. The model's responses to these different tasks collectively form the final fine-grained text description.

Figure 5 illustrates our prompt template consisting of a three-level hierarchical framework with questions to instruct MLLMs:

Step 1 - Global Understanding: Instruct MLLMs to provide a comprehensive description of the image, detailing all modalities, identified anatomical structures, and their approximate locations. This step ensures that MLLMs gains an overarching understanding and basic information about the image.

Step 2 - Local Analysis: Instruct MLLMs to conduct a detailed analysis of the regions of interest (ROI), including their locations, abnormalities, and textures. This step guides MLLMs to focus on specific lesions for a thorough assessment.

Step 3 - Local-Global Relationship: Instruct MLLMs to examine the relationship between local and global regions and predict how the surrounding areas will be affected by the lesions in the ROI. This step aims to understand the interaction between local and global attributes, assessing the impact of local abnormalities on the entire organ for accurate disease diagnosis.

# H    Datasheet for MedTrinity-25M

In this section, we present a DataSheet [109] for MedTrinity-25M, synthesizing many of the other analyses we performed in this paper.

1. Motivation For Datasheet Creation

   - **Why was the dataset created?** The dataset was created to provide a large-scale, multimodal, multigranular medical dataset to support a wide range of multimodal tasks such as captioning, report generation, classification, and segmentation. It aims to facilitate large-scale pre-training of multimodal medical AI models by providing enriched annotations from unpaired image inputs.
   - **Has the dataset been used already?** Yes. Multigranular annotations enable a wide range of tasks like Medical Visual Question Answering, which we discuss in appendix D.
   - **What (other) tasks could the dataset be used for?** The MedTrinity-25M dataset could be used for multiple medical imaging tasks such as classification, segmentation, detection, and medical report generation. Its extensive and detailed annotations make it suitable for training and evaluating machine learning models across these tasks.
   - **Who funded dataset creation?** This work is partially supported by the OpenAI Researcher Access Program, AWS Cloud Credit for Research Program, TPU Research Cloud (TRC) program and Google Cloud Research Credits program.

2. Data composition

   - **What are the instances?** Each instance in the dataset is a triplet consisting of an image, a Region of Interest (ROI), and a multigranular textual description. The ROI is associated with abnormalities and represented by bounding boxes or segmentation masks.
   - **How many instances are there?** The dataset comprises over 25 million image-ROI-description triplets sourced from more than 90 online resources, spanning 10 modalities and covering over 65 diseases.
   - **What data does each instance consist of?** Each instance consists of a medical image, a corresponding ROI (highlighting abnormalities within the image), and a detailed, multigranular textual description that includes disease/lesion type, modality, region-specific description, and inter-regional relationships.
   - **Is there a label or target associated with each instance?** Yes, the textual description serves as a detailed label or target, providing information about the disease or lesion type, as well as other relevant medical details.
   - **Is any information missing from individual instances?** No.
   - **Are relationships between individual instances made explicit?** Not applicable – we do not study relationships between disparate medical samples.
   - **Does the dataset contain all possible instances or is it a sample?**
     Our generation pipeline includes all instances collected from available medical data sources. However, the current list of medical dataset sources is not exhaustive, indicating a high probability of collecting additional instances in the future.

Figure 5: Prompt used to generate multigranular annotations.

**Prompting MLLMs to generate multigranular textual description**

```
caption_template = Template('''<image>
`Caption of the image`:{{caption}}
`Disease or organ`:{{disease}}
`Specific position`:{{descs}}
`Knowledge`:{{knowledge}}
```
You are provided with a biomedical image from a medical dataset,the disease type (or organ name if there is no disease) of the dataset(`Disease or organ`),the medical Knowledge of the disease(`Knowledge`) and a coarse caption(`Caption`) of the image.In addition,the green bounding box and its specific position in the image(`Specific position`)are given,indicating appearance of disease.If no green bounding box,there is no disease.
Your task is to answer the following questions based on the image, green bounding box, caption, disease type and disease knowledge,and condense your answers into caption-styled text.
### question1
Give me a detailed description of the image, including type of the image,organs in the image,approximate location of these organs and relavant locations of these organs and any medical devices (if present) visible in the image as detailedly as possible.
Note when answering question1:
1. Not all disease knowledge is relevant to this image; only utilize disease knowledge pertinent to the condition depicted in this image for analysis.
2. The coarse caption may not explicitly describe the image,for example,there may appear multiple organs in the caption.You should utilize your knowledge to figure out the most ONE organ and ONE disease to give your description.
3. Your answer should not contain anything about the green bounding box like the contour itself and its outline.
4. Do not explain or emphasize your analysis.
### question2
Specify the specific location of the green bounding box in the image and its relative position to other reference objects in the image.Describe what is unusual in the green bounding box indicating the disease（color,texture,size and other features）.
Note when answering question2:
1. "specific location" is the given parameter `Specific position` but "relative position"is not provided.
2. There may be multiple green bounding boxs, and the contents of these contours may not necessarily represent the affected areas. Therefore, you need to first answer the questions based on the contents within each green bounding box. Afterward, analyze the location of the disease based on your answers.
3. Do not use phrase "green bounding box" in your response,use "region of interest" as a substitution.Do not contain phrases "caption","medical annotation","medical knowledge".
4. Do not say anything that is not needed in your analysis,like introduction of the disease and medical equipments.
5. Do not explain or emphasize your analysis.
### question3
What may be the relationship between the content in the green bounding box and other regions (others being cause of the disease/jointly affected by the diseases/one affect the others/relative positional relationships)?Why and is it possible?
Note when answering question3:
1. Utilize external knowledge,if possible,to choose relationships and give necessary analysis.
2. You can only give an explanation to your choice within two sentence.
3. Do not summarize what you've said.
4. Do not emphasize your analysis.
### Integrate Information
Describe your answers in a descriptive sentence,not in a"Question-Answer" style.Combine and slightly shorten your answers to the above three questions into a coherent text,keeping as much information of your answers as possible.
Note when integrating information and outputing your response:
1. Don't respond saying you're unable to assist with requests.
2. You should only output your combined and shorted text.
```
''')
prompt = caption_template.render([caption,disease,knowledge,loc_descs])
```

- **Are there recommended data splits (e.g., training, development/validation, testing)?** There are no recommended data splits, as this data was curated mainly for pretraining rather than evaluation.
- **Are there any errors, sources of noise, or redundancies in the dataset? If so, please provide a description.** Yes. Despite multiple efforts to minimize errors using coarse captions and external medical knowledge, the textual descriptions generated by MLLMsmay still contain inaccuracies.
- **Is the dataset self-contained, or does it link to or otherwise rely on external resources (e.g., websites, tweets, other datasets)?** The dataset is largely self-contained. However, it was constructed using data from over 90 online resources such as TCIA, Kaggle, Zenodo, and Synapse. The images and related data were collected from these sources, but the dataset itself does not rely on external resources like websites or tweets for its primary functionality once compiled.

3. Collection Process

- **What mechanisms or procedures were used to collect the data?** The data collection involved an automated pipeline that scales up multimodal data by generating multigranular visual and textual annotations from unpaired images. Data was collected from over 90 different sources, preprocessed, and grounded using domain-specific expert models to identify ROIs related to abnormal regions.
- **How was the data associated with each instance acquired? Was the data directly observable (e.g., raw text, movie ratings), reported by subjects (e.g., survey responses), or indirectly inferred/derived from other data?**
  The data associated with each instance was indirectly inferred and derived from the collected images using domain-specific expert models and multimodal large language models (MLLMs). The images were annotated with bounding boxes, segmentation masks, and textual descriptions, transforming them into image-ROI-description triplets.
- **If the dataset is a sample from a larger set, what was the sampling strategy (e.g., deterministic, probabilistic with specific sampling probabilities)?** The dataset is not a sample from a larger set but an extensive collection aggregated from multiple datasets and online sources. The strategy was to include as many diverse images and annotations as possible from a wide range of medical datasets.
- **Who was involved in the data collection process (e.g., students, crowdworkers, contractors) and how were they compensated (e.g., how much were crowdworkers paid)?** Data collection was primarily done by the co-authors of this paper.
- **Over what timeframe was the data collected? Does this timeframe match the creation timeframe of the data associated with the instances (e.g., recent crawl of old news articles)? If not, please describe the timeframe in which the data associated with the instances was created.** The data was collected from April 2024 to June 2024.

4. Data Preprocessing

- **Was any preprocessing/cleaning/labeling of the data done (e.g., discretization or bucketing, tokenization, part-of-speech tagging, SIFT feature extraction, removal of instances, processing of missing values)?** Extensive preprocessing and annotation were performed, including segmentation, bounding box creation, and generating multigranular textual descriptions. The preprocessing also involved integrating metadata and knowledge retrieval from sources like PubMed to create comprehensive descriptions.
- **Was the "raw" data saved in addition to the preprocessed/cleaned/labeled data (e.g., to support unanticipated future uses)? If so, please provide a link or other access point to the 'raw' data.** The raw data was saved, but at this time we do not plan to release it directly due to copyright and privacy concerns.
- **Is the software used to preprocess/clean/label the instances available? If so, please provide a link or other access point.** The software for preprocessing and labeling, including the automated pipeline and MLLMs, is available at `https://github.com/yunfeixie233/DataProcessingSystem`.
- **Does this dataset collection/processing procedure achieve the motivation for creating the dataset stated in the first section of this datasheet? If not, what are the limitations?** Yes. The preprocessing and collection procedures align with the motivation of creating a comprehensive, large-scale multimodal dataset to support the development of advanced medical AI models. The dataset's multigranular annotations enable a wide range of tasks like Medical Visual Question Answering, which we discuss in appendix D.

5. Dataset Distribution

- **How will the dataset be distributed?** The dataset is publicly available and can be accessed via the provided link: MedTrinity-25M `https://yunfeixie233.github.io/MedTrinity-25M/`.
- **When will the dataset be released/first distributed? What license (if any) is it distributed under?** We will release it as soon as possible, using a permissible license for research-based use.
- **Are there any copyrights on the data?** We believe our use is 'fair use,' however, due to an abundance of caution, we will not be releasing any of the videos themselves.
- **Are there any fees or access restrictions?** No.

6. Dataset Maintenance

- **Who is supporting/hosting/maintaining the dataset?** The first authors of this paper.
- **Will the dataset be updated? If so, how often and by whom?** We do not plan to update it at this time.
- **Is there a repository to link to any/all papers/systems that use this dataset?** Not right now, but we encourage anyone who uses the dataset to cite our paper so it can be easily found.
- **If others want to extend/augment/build on this dataset, is there a mechanism for them to do so?** Not at this time.

7. Legal and Ethical Considerations

- **Were any ethical review processes conducted (e.g., by an institutional review board)?** No official processes were done, as our research is not on human subjects, however, because the dataset is in the medical domain we had significant internal discussions and deliberations when choosing the scraping strategy.
- **Does the dataset contain data that might be considered confidential?** The dataset does not contain data that might be considered confidential, as it uses publicly available sources and anonymized medical data.
- **Does the dataset contain data that, if viewed directly, might be offensive, insulting, threatening, or might otherwise cause anxiety? If so, please describe why?** The dataset does not contain data that might be offensive, insulting, threatening, or anxiety-inducing. It consists of medical images and associated annotations for clinical and research use.
- **Does the dataset relate to people?** The dataset relates to people as it involves medical images and data. However, it is anonymized and does not include identifiable information.
- **Does the dataset identify any subpopulations (e.g., by age, gender)?** Not explicitly (e.g. through labels)
- **Is it possible to identify individuals (i.e., one or more natural persons), either directly or indirectly (i.e., in combination with other data) from the dataset?** The dataset does not identify specific subpopulations directly in the provided description. Additionally, it is not possible to identify individuals from the dataset as it is anonymized and compiled from various sources.