# OpenReview forum: "MedTrinity-25M: A Large-scale Multimodal Dataset with Multigranular Annotations for Medicine"
_NeurIPS.cc/2024/Datasets_and_Benchmarks_Track — Submitted to NeurIPS 2024 Track Datasets and Benchmarks_

### Official Review · Reviewer_1AYs · 2024-07-24
**A very useful and comprehensive multi-modal medical dataset**

**Rating:** 6
**Confidence:** 4
**Correctness:** N/A
**Clarity:** N/A

**Review:**

High quality open-source work can be accepted, details shown above. But if there are some benchmark from the model side, that would be better.

**Strengths:**

Comprehensive and high-quality dataset, though there are existing multi-modal dataset. But this work has a comprehensive scale...

**Additional Feedback:**

N/A

**Documentation:**

N/A

**Opportunities For Improvement:**

Could benchmark the SOTA LLMs on this dataset (Though it is a huge work load, but happy to see it...

**Relation To Prior Work:**

N/A

**Summary And Contributions:**

The paper presents MedTrinity-25M, an expansive multimodal medical dataset featuring over 25 million images across 10 modalities, annotated for 65 diseases with both global textual information and detailed local annotations like bounding boxes and segmentation masks. This dataset innovates by introducing an automated pipeline capable of scaling multimodal data generation without requiring pre-existing text-image pairs, utilizing domain-specific models for ROI identification and leveraging multimodal LLMs for generating detailed annotations.

Contributions:

1. Development of MedTrinity-25M, a large-scale, richly annotated multimodal dataset for medicine, which includes comprehensive annotations for a wide variety of diseases and medical imaging modalities.
2. Introduction of an automated annotation pipeline that enhances data scalability by generating detailed visual and textual annotations without the need for paired text, which is a significant advancement over existing methods.
3. The dataset supports a broad array of multimodal and vision-centric tasks, facilitating the pre-training of advanced multimodal medical AI models, thereby laying the groundwork for future foundational models in the medical field.

---

> ### Author Rebuttal · Authors · 2024-08-17
>
> We thank your recognition of the quality of data and scalability of the pipeline. We address specific points below:
> 1. **Result of benchmarking the SOTA LLMs:**
>
> Thanks for your suggestion. We benchmarked several state-of-the-art MLLMs on this dataset using the same prompts to generate multigranular annotations, treating the annotations in our dataset as the ground truth for evaluation. To comprehensively assess these MLLMs, we employed common NLP metrics, including Recall, BLEU, and ROUGE.
>
> Due to time constraints, we uniformly sampled 5,000 instances from our dataset for evaluation. The results are presented below:
>
>
> | Model       | Recall   | BLEU-1  | BLEU-4  | METEOR  | ROUGE-L |
> |-------------|----------|---------|---------|---------|---------|
> | InternVL2   | 0.3858   | 0.3513  | 0.0894  | 0.2853  | 0.2729  |
> | LLaVA       | 0.2619   | 0.2897  | 0.0482  | 0.2069  | 0.2409  |
> | LLaVA-Med   | 0.2078   | 0.1404  | 0.0266  | 0.1457  | 0.2141  |
> | Mini-CPM    | 0.3145   | 0.2761  | 0.0692  | 0.2278  | 0.2557  |
> | Phi-3       | 0.3293   | 0.3165  | 0.0575  | 0.2450  | 0.2324  |
>
>
> These NLP metrics are comparatively low, highlighting the existing limitations of SOTA MLLMs in generating multigranular annotations.
> This indicates that our proposed dataset has the potential to enhance the multigranular annotation capabilities of SOTA MLLMs.

---

> ### Author Response · Authors · 2024-08-26
>
> Dear Reviewer 1AYs,
>
> We sincerely appreciate your review and have carefully considered each of your questions. Detailed responses are provided in the rebuttal, and we welcome any further questions or concerns you may have.

---

### Official Review · Reviewer_hsbA · 2024-07-26
**review of MedTrinity**

**Rating:** 5
**Confidence:** 4
**Correctness:** reasonable.
**Clarity:** fairly well.

**Review:**

The paper introduces a new benchmark, where the biggest contribution is probably the triplet of image, text, and ROI. The work can be potentially significant is used properly, however, as the data is annotated by MLLM, it seems undesired to use the dataset to evaluate other MLLMs. Thus, how to fully utilize the dataset might become a question.

**Strengths:**

+ the dataset is comprehensive.

+ the idea of introducing ROI as a new modality is important and interesting.

+ the method part is written fairly clear.

**Additional Feedback:**

N/A.

**Documentation:**

fair.

**Ethics:**

none noted.

**Limitations:**

The authors did not discuss limitations explicitly.

**Opportunities For Improvement:**

- the biggest question would be where the dataset is used in the future. Since the dataset itself is annotated with MLLM (with a database), it seems natural that a similar model would achieve the best performances over the benchmark. It might be necessary to demonstrate its practical relevance in training or evaluating other models.

- some expert validation might be necessary

- what will the ROI be in certain cases where it's indeed the global information that matters for that disease? or does such case exist in the benchmark?

**Relation To Prior Work:**

fair.

**Summary And Contributions:**

The paper introduces MedTrinity, one of the largest multi-modality dataset (image and text description) in the medical studies. In addition t the image and text, the authors also include the ROI as part of the data to help the study locate where the study should focus. The annotation is automatically conducted by a MLLM, which might have issues without human supervision.

---

> ### Author Rebuttal · Authors · 2024-08-17
>
> Thank you very much for your insightful comments and suggestions. We address specific points below:
>
> 1. **Performance of models trained on MedTrinity-25M:**
>
> In our experiment reported in supplymentary of the paper, we pretrained the LLaVA-Med++ using proposed MedTrinity-25M, and evaluated the performance of pretrained model on VQA datasets **instead of MedTrinity-25M**, including **VQA-RAD** and **SLAKE**.
> The performance improvement on LLaVA-Med++ suggests that the multigranular information in our dataset helps improving the overall medicine ability of multimodal models.
>
> Moreover, according to your suggestion, we provide results of the training other types of multimodal models including PubMedCLIP, MiniCPM-V-2.6 and InternVL2-8B (which are not ultilized to generate annotation in our dataset) in the global response.
> All of these models achieve performance improvement by pretraing on proposed MedTrinity-25M, suggesting the quailty and effectiveness of MedTrinity-25M. We will include these results in our revised paper.
>
>
> 2.  **Expert Validation:**
>
> Thanks for your suggestion. To assess the quality of the dataset, we distributed evaluation questionnaires to experts with medical backgrounds.
>
> Our dataset's structure is defined by five key attributes that characterize medical images: modality, structure detection, ROI analysis, lesion texture, and local-global relation. We provided both our generated annotations and expert annotations for the same samples and asked medical experts to evaluate the accuracy of each attribute to validate our approach. Each correct attribute received 1 point, while incorrect ones received 0 points, resulting in a score list for each sample, such as [1, 1, 1, 1, 0]. We then averaged these scores across all evaluated samples to obtain an overall expert validation score.
>
> We uniformly selected a subset of 200 samples for expert validation. The results we collected are presented below.
>
> | Score | Overall | Modality | Structure Detection | ROI Analysis | Lesion Texture | Local-Global Relation |
> |-------|---------|----------|---------------------|--------------|----------------|-----------------------|
> | Ours  | 0.85/1.00 | 1.00/1.00  | 0.88/1.00             | 0.88/1.00          | 0.69/1.00        | 0.81/1.00               |
>
> According to the medical expert validation, we received relatively high scores, confirming the validity of our dataset. We will include this medical expert validation in our revision paper.
>
> 3. **Example of ROI when global information matters:**
>
> Thanks very much for your suggestion.
> We acknowledge that global information can sometimes be crucial in disease diagnosis.
> However, in our dataset, for samples with disease, the ROIs will always highlight the lesion areas, including both necessary global and local information of the diseases.
> For example, in chest radiography, a comprehensive analysis of organs across the entire image, including both the left and right lungs, is often necessary to determine the disease, as demonstrated in the case provided in the attached PDF file.
> In such cases where information from the entire image is necessary,  the ROIs will encompass all regions associated with the disease, providing detailed analysis of each lesion area, including texture and patterns.
> This local analysis contributes to the overall diagnosis. By providing multigranular annotations that include both local and global information, our dataset facilitates aligning foundation models into the medical domain.

---

> ### Author Response · Authors · 2024-08-26
>
> Dear Reviewer hsbA,
>
> Thank you for your review. We have carefully addressed each of your questions in detail within the rebuttal and would appreciate any further feedback you may have.

---

### Official Review · Reviewer_VFKd · 2024-07-27
**MedTrinity-25M: A Large-scale Multimodal Dataset with Multigranular Annotations for Medicine**

**Rating:** 6
**Confidence:** 3
**Correctness:** The whole process is reasonable.
**Clarity:** The paper can describe the content cl…

**Review:**

**Pros:**

- The paper introduces MedTrinity-25M, a large-scale multimodal medical dataset with multigranular annotations.
- It develops an automated pipeline to scale up multimodal data using multimodal large language models (MLLMs) without relying on paired text descriptions.
- The paper builds a comprehensive knowledge base and uses Retrieval-Augmented Generation (RAG) to prompt multimodal large language models, generating multigranular visual and textual annotations enriched with knowledge based on regions of interest (ROIs).

**Cons:**

- The format of the paper has several issues. For example, the conclusion has a different style compared to other sections.
- The results of the model training on the proposed MedTrinity-25M dataset need to be included in the main paper. Although MedTrinity-25M appears to be reasonable and valuable, the evaluation of model training on the proposed dataset should be contained within the paper.

**Strengths:**

**Pros:**

- The paper introduces MedTrinity-25M, a large-scale multimodal medical dataset with multigranular annotations.
- It develops an automated pipeline to scale up multimodal data using multimodal large language models (MLLMs) without relying on paired text descriptions.
- The paper builds a comprehensive knowledge base and uses Retrieval-Augmented Generation (RAG) to prompt multimodal large language models, generating multigranular visual and textual annotations enriched with knowledge based on regions of interest (ROIs).

**Additional Feedback:**

The format of the paper has several issues.

**Documentation:**

Yes

**Ethics:**

There are no ethical concerns with the paper.

**Opportunities For Improvement:**

**Cons:**

- The format of the paper has several issues. For example, the conclusion has a different style compared to other sections.
- The results of the model training on the proposed MedTrinity-25M dataset need to be included in the main paper. Although MedTrinity-25M appears to be reasonable and valuable, the evaluation of model training on the proposed dataset should be contained within the paper.

**Relation To Prior Work:**

Yes

**Summary And Contributions:**

This paper introduces a large-scale multimodal dataset for medicine, comprising over 25 million image-ROI-description triplets sourced from more than 90 online resources.

---

> ### Author Rebuttal · Authors · 2024-08-17
>
> Thank you very much for your valuable comments. We address specific points below:
>
> 1. **Paper Formatting**:
>
> We have carefully revised our paper according to your suggestion, and addressed the inconsistency in style between the conclusion and other sections. In our updated version, the entire paper has been revised to maintain a consistent format and style.
>
> 2. **Results of the model training**:
>
> Thank you for your suggestion. The results of the model training on the proposed MedTrinity-25M dataset is illustrated in the global response. We will append these experiments into paper.

---

> ### Author Response · Authors · 2024-08-26
>
> Dear Reviewer VFKd,
>
> We sincerely appreciate your review. We have carefully considered each of your questions and provide detailed responses in the rebuttal. Please let us know if you have any further questions or concerns.

---

### Author Rebuttal · Authors · 2024-08-17

We thank all reviewers for acknowledging the contribution of this work and their constructive and insightful feedback. We are delighted that they appreciate the following: _''a large-scale multimodal medical dataset, ... generating multigranular visual and textual annotations enriched with knowledge based on ROIs.''_(**Reviewer VFKd**), _''dataset is comprehensive, ... idea of introducing ROI as a new modality is important and interesting.''_(**Reviewer hsbA**), _''supports a broad array of multimodal and vision-centric tasks, ... laying the groundwork for future foundational models.''_(**Reviewer 1AYs**).

The primary concerns raised in the paper include the formatting and section organization issues (**Reviewer VFKd**), the need of expert validation (**Reviewer hsbA**), and the need for experiment results of the model training (**Reviewer VFKd and Reviewer hsbA**) or benchmarking (**Reviewer 1AYs**) on the MedTrinity-25M.
Our responses to these questions and suggestions can be summarized as follows:
- **Additional experiments**: we provide the results of models training on MedTrinity-25M in this global response. Moreover, we demonstrate the results of benchmarking SOTA LLMs on MedTrinity-25M in response to Reviewer 1AYs.
- **Expert Validation**: we conduct expert validation for the quality of MedTrinity-25M. We report the results in reponse to Reviewer hsbA.
- **More example of ROI**: we provide example of ROI in cases where it's indeed the global information matters for the disease. The example is provided in the attached PDF file. we provide in-depth explanation of given example in response to Reviewer hsbA.

### Results of model training on MedTrinity-25M:

To verify dataset quality, we developed a new medical multimodal model named LLaVA-Med++, and compared its performance with and without pretraining using MedTrinity-25M.
The model was initially pretrained on a subset of MedTrinity-25M, then finetuned and evaluated on popular medical VQA benchmarks of VQA-RAD, SLAKE, and Path-VQA. Performance comparisons are shown in the table below.

The  w/ and w/o indicate models with and without pretraining on MedTrinity-25M, respectively.
The asterisk (*) indicates that some methods treat open-ended questions as classification tasks, potentially overestimating performance.
| Method                    | VQA-RAD      |              | SLAKE        |              | PathVQA      |              |
|---------------------------|--------------|--------------|--------------|--------------|--------------|--------------|
|                           | Open         | Closed       | Open         | Closed       | Open         | Closed       |
| GPT-4V [1]                | 39.5         | 78.9         | 33.6         | 43.6         | -            | -            |
| LLaVA [2]                 | 50.0         | 65.1         | 78.2         | 63.2         | 7.7         | 63.2         |
| LLaVA-Med [3]             | 55.5         | 66.5         | 70.6         | 54.5         | 35.9        | 89.2         |
| VL Encoder-Decoder [4]    | 71.5*        | 82.5         | -            | -            | 71.5*        | 85.6         |
| Prefix T. Medical LM [5]  | -            | -            | 84.3*        | 82.0         | 40.0*        | 87.0         |
| BiomedCLIP [6]            | 67.6*        | 79.8         | 82.1*        | 89.7         | -            | -            |
| M2I2 [7]                  | 66.5*        | 83.5         | 74.7*        | 91.1         | 36.3*        | 88.0         |
| LLaVA-Med++(Ours, w/o)    | 64.6         | 77.0         | 79.3         | 84.0         | 55.0        | 94.0         |
| **LLaVA-Med++(Ours, w/)** | **77.1 (+12.5)** | **86.0 (+9.0)** | **86.2 (+6.9)** | **89.3 (+5.3)** | **66.5 (+11.5)** | **99.0 (+5.0)** |
| PubMedCLIP [8] (w/o)      | 55.56*       | 79.34        | -            | -            | -            | -            |
| **PubMedCLIP [8] (w/)**   | **60.56\* (+5.0)** | **79.70 (+0.36)** | -            | -            | -            | -            |
| MiniCPM-V-2.6 [9] (w/o)   | 65.0         | 95.9         | 76.2         | 82.7         | -            | -            |
| **MiniCPM-V-2.6 [9] (w/)**| **74.4 (+9.4)** | **99.2 (+3.3)** | **77.0 (+0.8)** | **85.5 (+2.8)** | -            | -            |
| InternVL2-8B (w/o) [10]   | 38.2         | 76.2         | 61.7         | 77.8         | 16.8        | 86.4         |
| **InternVL2-8B (w/)** [10]| **40.7 (+2.5)** | **80.0 (+3.8)** | **66.4 (+4.7)** | **78.8 (+1.0)** | **23.6 (+6.8)** | **87.4 (+1.0)** |

By employing pretraining on MedTrinity-25M, LLaVA-Med++ achieves SOTA or comparable performance in all benchmarks, suggesting the quality of multigranular MedTrinity-25M.
Additionally, we pretrained multimodal models such as PubMedCLIP, MiniCPM-V-2.6 and InternVL2-8B. The results demonstrated in table above suggest that pretraining on MedTrinity-25M leads to significant performance boost across various models.
This finding highlights the potential of MedTrinity-25M in developing powerful medical foundation models.

### References

[1] Achiam, et al. "Gpt-4 technical report."

[2] Liu, et al. "Visual instruction tuning."

[3] Li, et al. "Llava-med: Training a large language-and-vision assistant for biomedicine in one day."

[4] Bazi, et al. "Vision–language model for visual question answering in medical imagery."

[5] Van, et al. "Open-ended medical visual question answering through prefix tuning of language models."

[6] Zhang, et al. "BiomedCLIP: a multimodal biomedical foundation model pretrained from fifteen million scientific image-text pairs."

[7] Li, et al. "Self-supervised vision-language pretraining for medial visual question answering."

[8] Eslami, et al. "Pubmedclip: How much does clip benefit visual question answering in the medical domain?."

[9] Yao, et al. "Minicpm-v: A gpt-4v level mllm on your phone."

[10] Chen, et al. "Internvl: Scaling up vision foundation models and aligning for generic visual-linguistic tasks."

---

### Author Response · Authors · 2024-09-05
**Follow-up Regarding Reviewers Feedback During Discussion Phase**

Dear All Reviewers,

We hope this message finds you well. We're following up on our paper (ID: 346).

The discussion phase has ended, but we haven't received any response to our rebuttal addressing your questions. We would greatly appreciate any feedback you can provide on our rebuttal.

Thank you for your time and consideration.

Best regards,

The Authors

---

### Decision · Program_Chairs · 2024-09-26

**Decision:**

Reject

**Comment:**

Since the dataset was generated partially with the help of a multimodal large language model (MLLM), the main concerns from the reviewers include the quality of the dataset and whether the dataset can help further improve the state-of-the-art MLLM. In the rebuttal, the authors provided additional experimental results to address the above concerns. Unfortunately, all three reviewers did not participate in the discussion nor updated the scores. Although the additional results are convincing enough to dismiss most of the concerns, the score is not high enough to justify acceptance.